# Detecting topological variations of DNA at single-molecule level

Ke Liu [1], Chao Pan[2], Alexandre Kuhn[3,5], Adrian Pascal Nievergelt [4], Georg E. Fantner [4],
Olgica Milenkovic[2] & Aleksandra Radenovic [1]

In addition to their use in DNA sequencing, ultrathin nanopore membranes have potential applications in detecting topological variations in deoxyribonucleic acid (DNA). This is due to the fact that when topologically edited DNA molecules, driven by electrophoretic forces, translocate through a narrow orifice, transient residings of edited segments inside the orifice modulate the ionic flow. Here we utilize two programmable barcoding methods based on base-pairing, namely forming a gap in dsDNA and creating protrusion sites in ssDNA for generating a hybrid DNA complex. We integrate a discriminative noise analysis for ds and ss DNA topologies into the threshold detection, resulting in improved multi-level signal detection and consequent extraction of reliable information about topological variations. Moreover, the positional information of the barcode along the template sequence can be determined unambiguously. All methods may be further modified to detect nicks in DNA, and thereby detect DNA damage and repair sites.

---

[1] Laboratory of Nanoscale Biology, Institute of Bioengineering, School of Medicine, EPFL, 1015 Lausanne, Switzerland. [2] Electrical and Computer Engineering, University of Illinois at Urbana-Champaign, Urbana, IL 61801, USA. [3] Laboratory of Molecular Biotechnology, Institute of Biotechnology, University of Lausanne, 1015 Lausanne, Switzerland. [4] Laboratory for Bio- and Nano-Instrumentation, Institute of Bioengineering and Institute of Microengineering, School of Engineering, EPFL, 1015 Lausanne, Switzerland. [5] Present address: Selexis SA, 1228 Geneva, Switzerland. Correspondence and requests for materials should be addressed to K.L. (email: ke.liu@epfl.ch) or to A.R. (email: aleksandra.radenovic@epfl.ch)

Nanopore sensing is an emerging single-molecule DNA or RNA sequencing technology which holds great promise for enabling long readout signals[1–3]. There are two types of nanopores in current use: biological nanopores (α-hemolysin[4], MspA[5]) and solid-state nanopores (SiN$_x$[6–8], 2D materials[9–14], and glass[15]). Biological nanopores have several unique properties that enable DNA strand sequencing based only on temporal ionic current signals, thereby challenging the current state-of-art next-generation sequencing techniques that exploit fluorescent labeling and optical detection.

The two types of nanopores differ in several aspects: First, biological pores have well-defined pore geometries with pre-scribed atomic precision. The narrow constriction (1.4 nm for α-hemolysin and 1.2 nm for MspA) facilitates the sequential differentiation of individual nucleotides along the strand. Second, the incorporation of a so-called "molecular motor" onto the pore mouth can ratchet single-strand DNA in a base-by-base manner. However, this also limits the sequencing speed to 450 nt/s[16]. Third, the lipid bilayer used to host biological pores can be chosen to have a low dielectric constant for low-noise sensing. Despite these advantageous features, biological pores also have some intrinsic drawbacks: The fragility of the lipid bilayer that supports the pore; the technical difficulties encountered to graft biological pores into a large-scale array (more than 1,000,000 pores/cm$^2$); and most importantly, the structural drawback related to the long β-barrel in α-hemolysin which "dilutes" the ionic current signal of a single nucleotide from the narrow constriction. Thus, a single-nucleotide resolution is hard to expect from such a pore structure.

Our work based on atomically thin MoS$_2$ nanopore represents a development in DNA sequencing in favor of solid-state nanopores[17]. Single-layer MoS$_2$ has a thickness of 6.5 Å[13,18] equal to twice the distance between two bases in dsDNA. Several theoretical studies have predicted that MoS$_2$ has a superior performance in base reading due to its unique electronic properties[19–21]. In realistic strand sequencing applications, the spatial uncertainty created when the strand translocates through the nanopore is vital to determining the location information and in de novo assembly of the target sequence. Spatial resolution in traditional SiN$_x$ nanopores is limited to more than ten bases due to its thickness. Nanopore data reduce to a time-resolved current signal. Hence, the accuracy of conversion from a temporal signal into a spatial sequence depends on the homogeneity of the translocation velocity. So far, the homogeneity of the translocation velocity remains a major point of contention in the literature. Singer et al.[22] used synthetic peptide nucleic acid probes attached to a target viral gene and showed good localization certainty using translocation data. Plesa et al.[23] observed a large velocity fluctuation related to the reduced drag force exerted on the untranslocated part of the DNA coil while long DNA unfolds to translocate through the nanopore. Bell et al.[24] exploited a digital encoding method to study the long-range arrangement of targeted sites along DNA strands. As a result, this raises the issue of the uncertainty in resolving the localized information of a feature along the DNA strand. Before addressing this question, one has to first distinguish real translocation events from rejection events. Due to the expected high entropic barrier at the nanopore entrance, the probability of true translocation is not expected to be even close to one. For intact dsDNA, the wide distribution of dwell time and conformational folding complicate the interpretation of translocation data.

## Results

**ds–ss–ds DNA complexes**. In our experiment, we designed ds–ss–ds DNA complexes by annealing base-paired matched ssDNA to a template, and probed such a complex using the standard SiN$_x$ nanopore. To assess the readout success rate and velocity profile of rigid DNA segments through a small nanopore (~3 nm), we designed two identical dsDNA segments with approximately one Kuhn length (~50 nm), and linked them by a soft ssDNA segment. The scheme is sketched (Fig. 1a), where the ds–ss–ds DNA complex translocates through a SiN$_x$ nanopore in the presence of an electrical field. A small nanopore was created by a focused electron beam on a locally thinned SiN$_x$ membrane (Fig. 1b, c). A good control over the nanopore size is critical for all subsequent nanopore experiments (Supplementary Fig. 1). Ideally, the nanopore size should be around 3 nm in diameter, which barely allows the translocation of the complex. To verify the yield of the formed constructs, we used atomic force microscopy (AFM) to image the individual molecules in physiological conditions to obtain relevant height information. Figure 1d shows an example of a 1 μm × 1 μm scanned surface with more than 20 pieces of our DNA complexes. The overall yield of forming ds–ss–ds complex is more than 80%, a result obtained after analyzing a number of such scans. AFM gives a direct statistic based on counting single molecules. A zoomed-in image of one such complex (marked by the white dashed rectangle in Fig. 1d, e) represents the three segments being designed. Due to the low imaging forces of force–distance-based AFM in liquids, the measured height profile is close to the nominal height profile for single-stranded and double-stranded DNA. From the cross-sectional profile (Fig. 1e), one can easily distinguish two ds segments from the ss segment. After characterizing the constructs, we performed nanopore single-molecule translocation experiments. The hypothetical current trace with two discrete levels captures the structure of our complex (Fig. 1f).

During the nanopore translocation experiments, the complex was fed through the cis-side and a positive bias was applied on the trans-side with a pair of Ag/AgCl electrodes. Four molar LiCl were used to enhance the ionic signal and reduce the velocity[25]. More details about the approach can be found in the Methods section. According to typical events observed during such experiments (Fig. 2a), we have classified the events into the following groups: "212" type events, corresponding to complete translocation; other types, such as "2" and "21", that may be attributed to partial translocation. We used a classical conductance blockage model to compute the pore size and the nanopore length (see Supplementary Discussion 1 and Supplementary Table 1). Assuming two levels induced by ss segment and ds segment, we obtained a good agreement for the nanopore size and the nanopore length. As shown in the scatter plot of collected events, "212" type represent a small fraction of the total number of events. We used standard clustering algorithms to classify all types of events. Three major clusters were identified, and plotted using blue, red, and green circles. Only 10% of the events were assigned to the "212". The current blockages induced by ssDNA and dsDNA are 0.8 ± 0.15 and 1.5 ± 0.18 nA, at 400 mV, respectively. It is interesting that the mean dwell time of "2" type events coincide with the mean dwell time of the first level of "212" or "21" type events. So, we speculate that the "2" and "21" events are partial translocations. In our experiments, most of the spike-like events include a single level, indicating that only the first dsDNA segment entered the pore. This finding is striking and it indicates a high entropic barrier for a complete translocation in such a small nanopore. Partial translocatons have also been observed in biological nanopores[26–28], as well in the context of narrow solid-state nanopores[29,30]. However, most solid-state nanopores are relatively big, facilitating complete translocation. In contrast, for narrow nanopores (~3 nm), the entropic barrier plays a vital role. We used a barrier model to understand molecular transport cross a nanopore in our previous

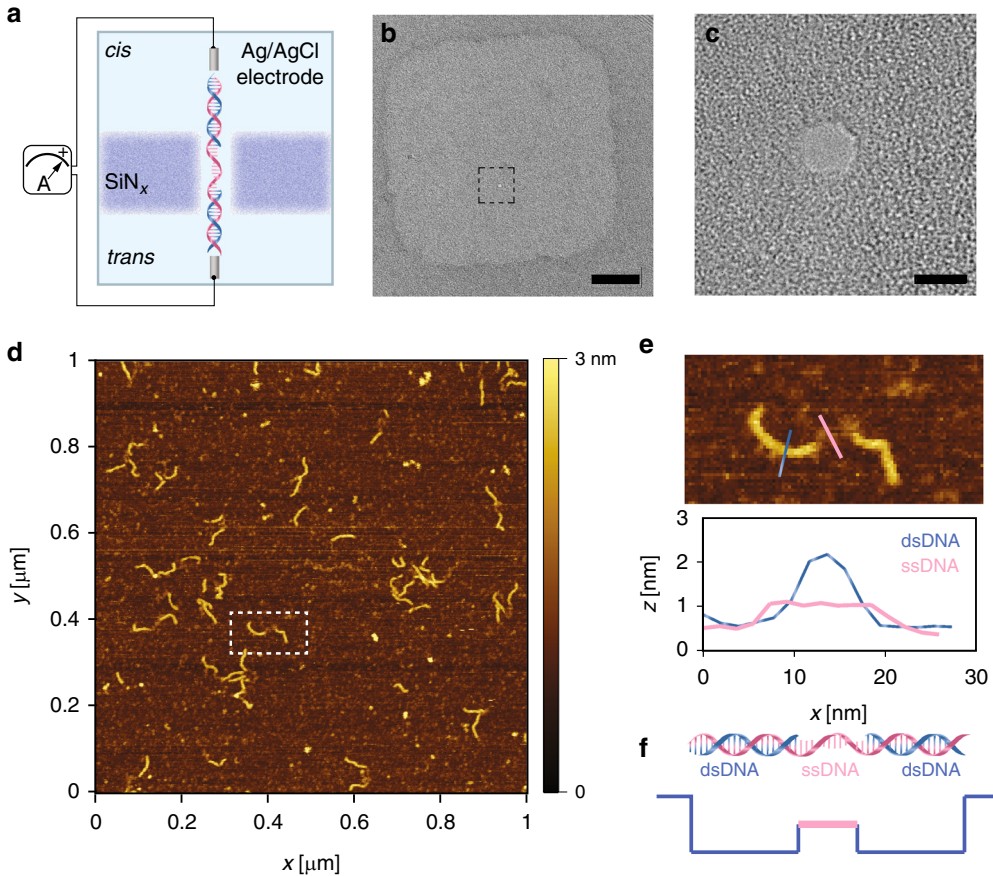

**Fig. 1 a** A schematic illustration of a nanopore membrane, where a programmed DNA passes through a nanopore in the $SiN_x$ membrane. The device is mounted between two reservoirs filled with a saline solution and connected to a current-to-voltage amplifier to induce a transmembrane bias across the membrane. **b** A low-magnification TEM bright-field image shows the locally thinned region on the $SiN_x$ membrane. The nanopore is marked by the black square. Scale bar is 50 nm. **c** A high-magnification TEM bright-field image presents a single 2.5 nm nanopore, which is slightly larger than the dsDNA cross-section (2.2 nm). Scale bar is 2 nm. **d** An overview AFM image of the programmed ds–ss–ds DNA complex on Mica under physiological conditions. About 20 molecules can be examined by one scan. **e** A zoom-in image of one hybrid marked by the white square in **d**. The height profile captures the hybrid nature of the compound. **f** The hypothesized current signal in the nanopore based on the DNA structure. Source data used to generate this figure can be found in the Source Data file

publication[17]. The probability of a complete translocation has an exponential dependence on the barrier energy.

Another important question that can be studied in our setting is the intramolecular velocity variation in "212" events (i.e., complete translocation). A large variation has been observed for the relative ratio of dwell time of the three segments (Fig. 2b, left). We used the relative ratio in dwell time to describe the variation for an average of 31 "212" events. In this case, we observed fractional dwell times, assuming total dwell time to be one, of $0.43 \pm 0.12$, $0.25 \pm 0.12$, and $0.32 \pm 0.08$ for each of the three segments, respectively (Fig. 2b, right). We used a reference map that assumes constant velocity and compared it to the experimental observations. The first ds segment has the lowest translocation speed among the three segments. The middle ss segment has the largest relative variation due to its softness. The last ds segment has a faster translocation speed than the first ds segment.

**Discriminative noise analysis**. To better understand the nature and variations of the two current levels, we focused on devising new methods for discriminating the two DNA topologies based on the empirical distribution and the power spectral density (PSD) of the noise associated with signals generated by no-DNA

(Level 0), ssDNA (Level 1), and dsDNA (Level 2). Noise analysis can facilitate the event detection for the "212" pattern and reveal the physical nature of the two different types of translocations (Fig. 3a). In order to reliably detect "212" events, one needs to accurately discriminate between transitions in the Level 1 and Level 2 phases. We used two transition prediction schemes: One in the time domain, based on simple Gaussian mixture model (GMM) fitting, and another one in the frequency domain, based on flicker and thermal noise modeling (see Supplementary Discussion 2). We start by considering the simplest noise model for Levels 1 and 2, in which we subtract the empirical average from the signals and then fit a probability distribution in the time domain for the remaining noise signal. Unfortunately, a Gaussian distribution cannot be used to model the noise present in Level 1 and Level 2 signals as it does not pass the Anderson–Darling test[31]. The histograms of those two signals exhibits a high degree of asymmetry (Supplementary Fig. 2), which may be intuitively justified as follows. When an ssDNA segment of a certain length enters the pore, one out of four possible nucleotides may modulate the ion current at any point of time, so that the histogram of the unknown signal fits that of a 4-mixture model. Similarly, when a dsDNA segment enters the pore, only two pairs of nucleotides are possible, causing another output 2-mixture distribution. Those two mixture models should be clearly

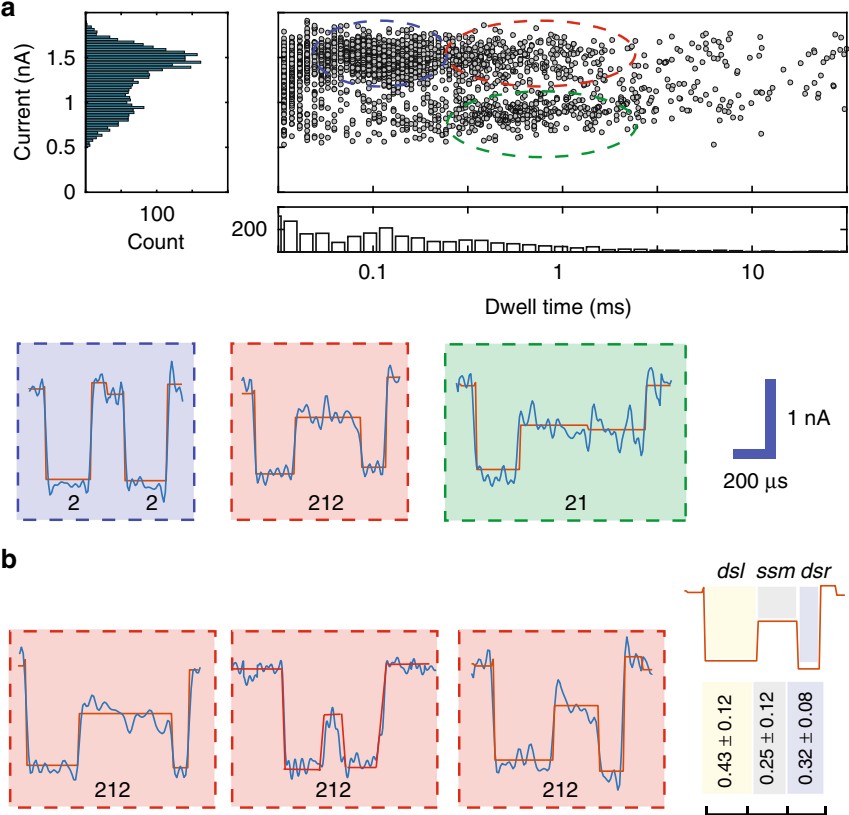

**Fig. 2 a** Scatter plot of a translocating ds–ss–ds DNA complex in a nanopore. Three clusters (blue, red, and green) denote the events that can be seen from the plot. Representative events for each cluster are listed below and denoted as "2", "212", and "21", respectively. A routine CUSUM[47] method is used to fit the multi-levels. **b** For "212" type events, a large intramolecular velocity variation is observed. The propositional average translocating time for each segment is calculated statistically (using 31 molecules). Source data used to generate this figure can be found in the Source Data file

distinguishable, as each nucleotide and each Watson–Crick pairing of nucleotides gives rise to different noise readings, both in terms of ground levels and noise variance. Hence, we fit the noise of nine typical "212" events from observed data by a GMM. A detailed description of the Anderson–Darling test and the definition of a GMM are provided (see Supplementary Discussion 2).

The quality of the GMM approximation improves with the number of components in the mixtures model, and with four components we observe that the weighted distribution distance

$$D = n \int_{-\infty}^{\infty} (F_n(x) - F(x))^2 \omega(x) \mathrm{d}F(x), \qquad (1)$$

between the hypothesized GMM cumulative distribution function and the empirical cumulative distribution function lies within the tolerance boundaries of our hypothesis test.

Here, $n$ stands for the number of data points, while $\omega(x)$, the weight function, equals

$$\omega(x) = [F(x)(1 - F(x))]^{-1}. \qquad (2)$$

As already described, a four-component model is also intuitively justified by the fact that each signal is modulated not only by the dsDNA or ssDNA topology, but also by the actual symbol composition of the DNA strands (involving the four nucleotides). A four-component GMM allows for identifying typical differences in the parameter of the noise associated with Level 0, Level 1, and Level 2 signals (Fig. 3b, c), which can be used to discriminate between them (here, the Level 0 signal

corresponds to the absence of DNA inputs). For example, one discriminative feature is the significantly larger number of small current values in Level 2 noise as opposed to Level 1 noise. Furthermore, Level 2 noise tends to exhibit steeper increases or decreases in the signal values when compared to Level 1.

Noise in solid-state nanopores has also been studied in the frequency domain, in which relevant noise components include flicker noise, dielectric noise, and capacitive noise[32–34]. It is now well-established that solid-state nanopore signals generated in the base current (Level 0) regime have a significant flicker noise component in the low-frequency (LF) domain, and that the noise characteristics change based on the content of the pore. Of special interest are the PSD models described in the earlier publications[32–34], which assert that the noise PSD may be described by Eq. (3), where $f$ denotes the noise frequency and where $a_1$, $a_2$, $a_3$, and $a_4$ are the corresponding coefficients in the expansion. LF flicker noise is mostly captured by the first component, where $\beta > 0$, while white thermal noise is captured via the term $a_2$, dielectric noise via $a_3 f$, and capacitive noise via $a_4 f^2$. Unlike all previous noise modeling approaches, we try to fit the noise model not just to a Level 0 signal, but to Level 1 and Level 2 signals as well. Figure 3d shows that, indeed, there exist observable differences in the noise spectra associated with different current levels in LF part. The frequency components for $f < 5$ kHz are consistent with the published results, and they follow a $1/f$ law, which is easily fitted by a single term $a_1/f^\beta$, different levels have different values for the parameters $a_1$ and $\beta$ (Table 1). However, for Level 1 and Level 2 noise with $f \leq 200$ Hz, our measurements are more consistent with white thermal noise. But this should still be a distinguishing feature of Level 0 noise on one side and Level 1

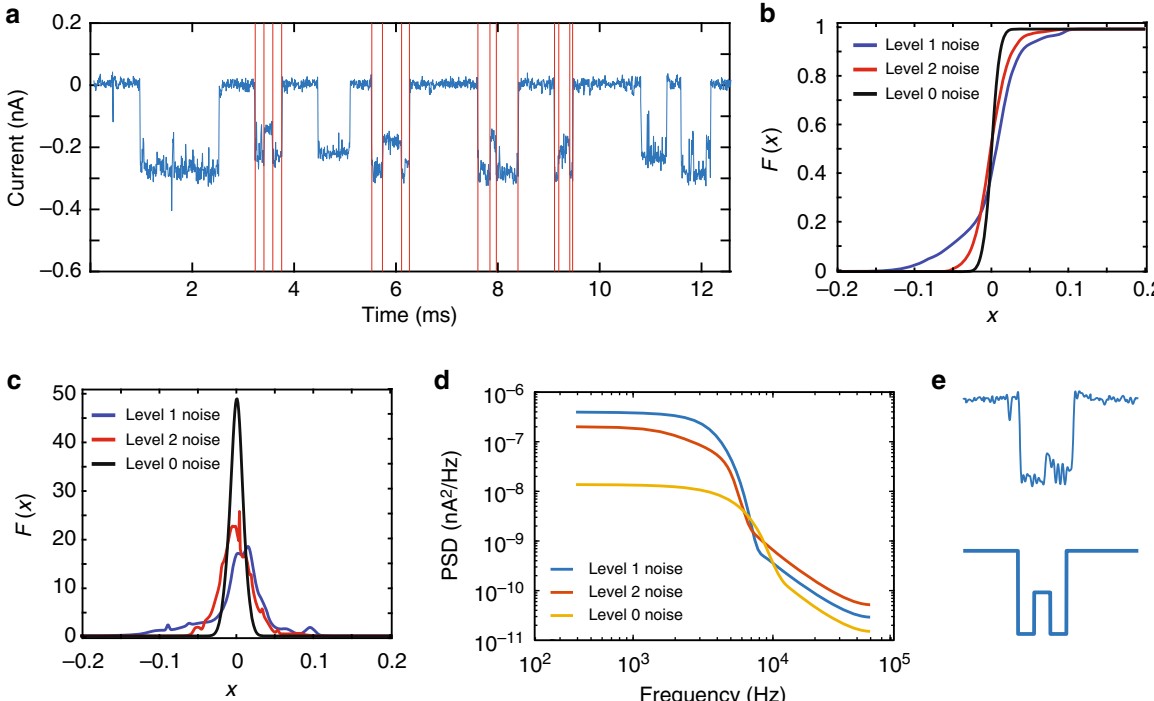

**Fig. 3 a** Typical curent trace, with current modulations induced by complexes translocating through a nanopore. Red lines indicate segments of Level 1 and Level 2 in the events. **b** Cumulative density functions (CDFs) of Level 0, Level 1, and Level 2 noise using a 4-component Guassian mixture model (GMM). **c** Probability density functions (PDFs) of Level 0, Level 1, and Level 2 noise using a 4-component Guassian mixture model (GMM). **d** Power spectral density (PSD) of Level 0, Level 1, and Level 2 noise using standard pwelch methods. **e** A sample "212" event, in which the Level 1 signal is hard to accurately discriminate without the noise spectrum analysis. Source data used to generate this figure can be found in the Source Data file

| Table 1 Fitted noise parameters for frequencies $f < 5$ kHz | | |
|---|---|---|
| | $a_1$ (with a 95% confidence bound) | $\beta$ (with a 95% confidence bound) |
| Level 0 noise | $4.14 \times 10^{-5}$ ($1.30 \times 10^{-5}$, $6.98 \times 10^{-5}$) | 0.202 (0.110, 0.294) |
| Level 1 noise | $3.70 \times 10^{-3}$ ($1.90 \times 10^{-3}$, $9.30 \times 10^{-3}$) | 0.380 (0.170, 0.591) |
| Level 2 noise | $3.15 \times 10^{-3}$ ($9.33 \times 10^{-4}$, $7.24 \times 10^{-3}$) | 0.473 (0.289, 0.656) |

and 2 noise on the other side. To perform spectral fitting, we used Welch's method for estimating the PSD (see Supplementary Discussion 2).

$$ S = a_1 \frac{1}{f^\beta} + a_2 + a_3 f + a_4 f^2. \qquad (3) $$

The fitted parameters are listed in Table 1.

We conclude the analysis by pointing out that a recent trend in nanopore signal processing is to use the Hilbert–Huang Transform (HHT)[35]. HHT is a useful empirical tool for analyzing nanopore traces as it is designed to work for signals that are nonstationary and nonlinear (see Supplementary Discussion 2). PSD analysis is suitable for topological signal detection, as the information-bearing signal in the studied setting has significantly simpler spectral discriminators than the one encountered when performing DNA bases detection or estimation. Still, HHT remains a useful tool that self-adapts to nonstationary and nonlinear nanopore signals. To illustrate the point, we performed the HHT analysis, and one of the findings is shown (Supplementary Figs 3 and 4). Note that Level 1 signals tend to have larger energy in the given frequency range than level 0 and 2 signals, and consequently, energy may be used as a means to

discriminate the events. Better results may be expected for a larger number of events, as our analysis only makes use of 40 samples.

**Protruded ssDNA in MoS₂ nanopores**. In solid-state nanopores based on SiN$_x$, due to the material's intrinsic thickness, its spatial resolution is limited to 5 nm[36], which converts into 15 bases. To be able to sense a small feature, the nanopore volume should be as small as possible to enable good sensitivity. In order to pass the barcoded hybrid, the nanopore orifice must be slightly bigger than the cross-section of the hybrid, which is ~2 nm. And in the meantime, the nanopore membrane should be as thin as possible. To take the advantage of a thin MoS₂ nanopore, we used short oligos (22–30mers or 7–9 nm) carrying a small barcoded single (or multiple) feature sequence to calibrate the velocity profile based on the temporal position of the secondary blocking strand on the template.

Figure 4a illustrates the basic experimental setup, in which a single-layer MoS₂ nanopore membrane is sandwiched between two reservoirs filled with ionic buffer and equipped with a pair of Ag/AgCl electrodes to apply a transmembrane potential. A barcoded ssDNA translocates through a nanometric pore on the MoS₂ membrane under electrophoretic force. A single-layer MoS₂ is suspended over a ~50 nm opening on an SiN$_x$ supporting membrane to have a confined region for nanopore formation

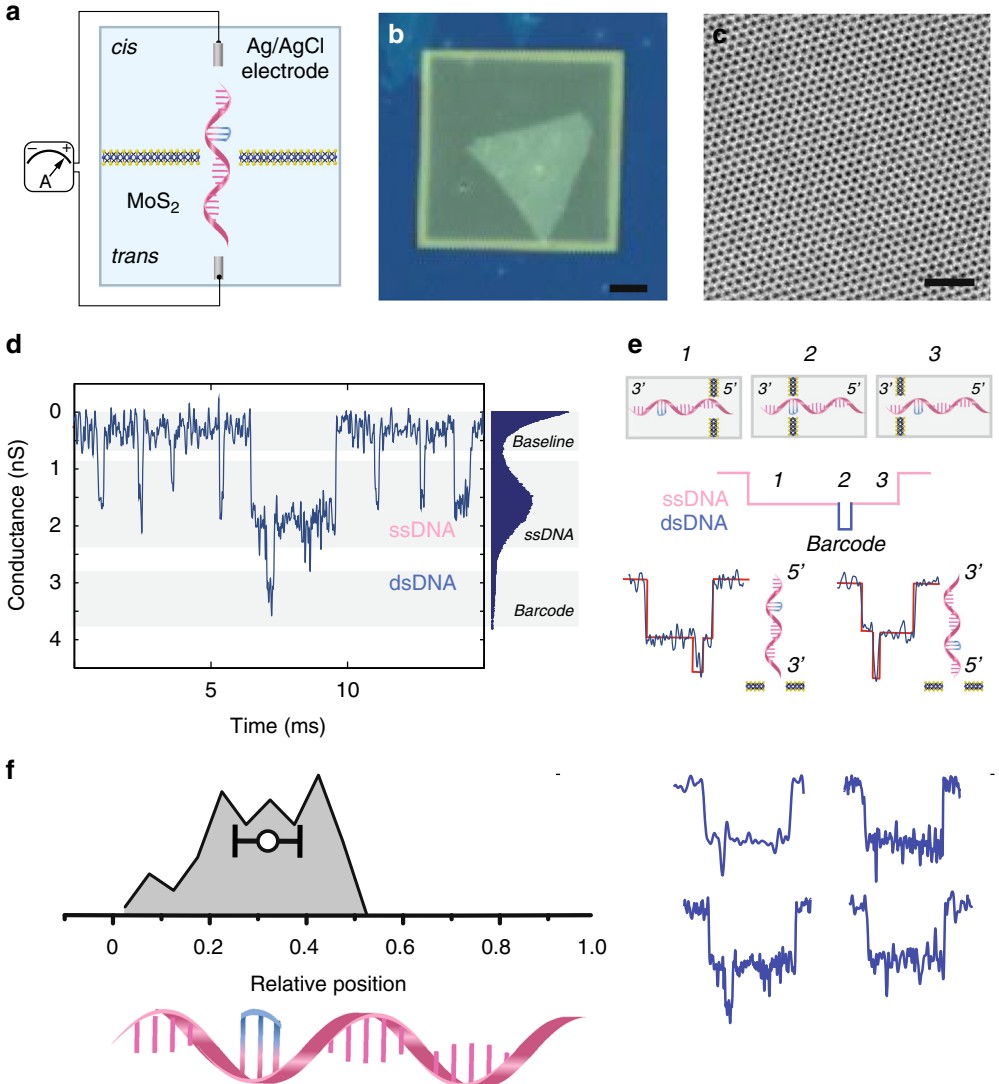

**Fig. 4 a** A schematic illustration of a MoS$_2$ nanopore membrane, where a barcoded DNA passes through a MoS$_2$ nanopore supported on the SiN$_x$ membrane. The device is mounted between two reservoirs filled with saline solution connected to a current amplifier to apply a transmembrane bias across the membrane. **b** A bright-field optical image of as-transferred single-layer MoS$_2$ over the supporting SiN$_x$ membrane. Scale bar is 10 μm. **c** An HRTEM image (high-resolution transmission electron microscopy) of the pristine MoS$_2$ lattice of size 12 nm × 12 nm. Scale bar is 2 nm. **d** A typical current trace illustrates one barcoded DNA signal together with other seven template ssDNA signals. The all-points histogram shows three discrete levels for an open pore baseline, ssDNA, and barcode, respectively. **e** A cartoon illustration of the translocation trajectory through the nanopore. The hypothesised current signal in nanopore shows the discrete 3-step process. In the experiments, two scenarios can be detected, one in which we have 3′ entering and another one in which we have 5′ entering. **f** A histogram of the relative temporal positions of the secondary-level centers (104 events) and its correlation to the longitudinal positions of the barcode in the sequence. Normalized events are dispalyed on the right. Source data used to generate this figure can be found in the Source Data file

while maintaining mechanical stability and reducing 1/$f$ noise (Fig. 4b). The suspended part of the MoS$_2$ layer is carefully characterized by transmission electron microscope (TEM). Figure 4c shows crystalline features of the MoS$_2$ lattice, indicating the good quality of the material. Previously, we have developed an electrochemical reaction (ECR) method[37] to create a single nanopore on MoS$_2$ substrates with sub-nm precision. The current–voltage (IV) characteristics (Supplementary Fig. 5) are before (black curve) and after (red curve) the formation of a 3 nm pore. The pore size can be calculated from a widely accepted pore size versus conductance model without using the time-consuming TEM procedure to determine the pore size[38,39].

The sequence 3′-GACTAGGGTGACAATCTACCAC-5′ was used as the template ssDNA with the barcode 5′ (Propynyl-C)$_3$3′ binding to the complementary sequence on the template. Here,

we used a melting-temperature ($T_m$) enhanced base, propynyl-modified cytosine, to improve the stability of the complex. The translocation measurements are done at 4 °C to keep the barcode hybridized to the template. Figure 4d shows a continuous trace with a high event frequency of 20 Hz. Translocation events appear in the trace as downward spikes due to the transient residing of single DNA molecules inside the nanopore. When inspected carefully, two types of events are observed: shallow and fast ssDNA translocation events, and multi-level and slow barcoded ssDNA translocation events. The all-points histogram (Fig. 4d right) indicates the level of the dsDNA barcoded part equals ~3.2 nS, and is much higher than the level of 1.8 nS observed for ssDNA. We used a threshold of 2 nS in blockage conductance to categorize barcoded events from ssDNA events. In the scatter plot (Supplementary Fig. 6), and quite surprisingly,

barcoded events have much larger dwell time compared to ssDNA. This is probably due to the friction between the barcode and the edge of the $MoS_2$ nanopore. Electrochemically etched $MoS_2$ nanopores have oxide protruding around the pore fringe, which has various possibilities of functionalization[19], and thus can further slow down the translocation. Most barcoded translocation events take place within milliseconds, which translates into a velocity range of 2000–20,000 nt/s. The fact that this speed in solid-state nanopores is much faster than that in biological nanopores potentially allows a much higher throughput in the former case. With the state-of-art current amplifier (Chimera Instruments, Inc.) with a bandwidth of up to 1 MHz, sufficiently many points are acquired for each nucleotide for its identity to be resolved.

Figure 4e illustrates a schematic translocation event when a barcoded short ssDNA passes through the $MoS_2$ nanopore. The barcode signature is represented as the secondary blockage event in the current signal. In order to have a sufficiently high temporal resolution, we used a salt gradient condition (cis with low salt concentration, trans with high salt concentration) to facilitate the capture of DNA segments and elongate the translocation time[40]. Such a complex has an asymmetric structure, so two possible orientations might happen during the translocation, as sketched. We did not observe any directional preference. Both orientations have almost the same chance to enter the pore. As shown in the scatter plot (Supplementary Fig. 7), the 5′ end has a 52% chance of entering, while the 3′ end has a 48% chance of entering. For the former case, the average dwell time is 3 ± 1 ms, and for the latter 1 ± 0.5 ms. This finding somewhat contradicts prior findings regarding the preferred entering direction in biological nanopores[41]. Both ends have an equal probability to find the pore entrance, but experience different friction during translocation. In a 3 nm $MoS_2$ nanopore, the temporal information of the barcode on the template strand can be clearly seen in the translocation event. The barcode is only a 1 nm feature, which surprisingly can be seen by the pulse resistive technique. This also indicates that the total volume of the feature is comparable to the sensing volume of the nanopore orifice. This is, to the best of our knowledge, the first time that a sub-1 nm spatial resolution has been demonstrated experimentally with solid-state nanopores. This is mainly due to the usage of a thin $MoS_2$ monolayer. The reason for not being able to detect features smaller than 1 nm is the temperature required to stabilize the hybrid. Better hybridization protocols will be explored in the future to enable single-base resolution detection.

We normalized all events to have the same dwell time in arbitrary units due to the variation in dwell time, and also mirrored 5′ entering events to 3′ entering events (Fig. 4f). For each event, we found the center of the barcoded level and its relative temporal position in the whole event. We used a histogram to plot all the relative temporal positions of the secondary-level centers and found a good correlation to the longitudinal positions of the barcode in the sequence. This result is different from the uncertainty of localized information observed in long DNA, as we can assume that the translocation velocity of short oligomers is constant during translocation. The Zimm relaxation time[42] of such a short oligomer is on the same order as its translocation time. In this case, the whole process can be considered to be a slow translocation. This can further explain the good correlation between the temporal position and the longitudinal position on the sequence.

## Discussion
Our findings show that the temporal position of the barcode properly correlates with its longitudinal position on the template sequence. This correlation implies that the translocation velocity of short oligos can be considered to be constant[43]. Short circulating miRNA (~26 nt) or non-coding DNA (~100 bps) have recently drawn great attention due to the facts that they play an important role in regulating gene expression, and they serve as biomarkers for certain diseases. Our nanopore-based platform can therefore also be used as a single-molecule sensing tool that allows for rapid detection and identification of biomarkers, eliminating false-positive results caused by error-prone amplification in traditional PCR diagnostic methods. It may also have potential applications in aligning unknown sequences with multiple known barcodes. This approach can also be extended to multiple-barcode schemes. For example, several barcodes can be detected simultaneously within a single translocation event, which has been demonstrated in the protein and aptamer complex[44]. The reading strategy can be further extended to either multiple reads with a single barcode per read or a single read with multiple barcodes. One technical obstacle is the variation in nanopore fabrication. We observe large pore-to-pore difference in dwell time and current. Noise analysis and pattern recognition could be possible solutions to increase the fidelity of the method reported.

## Methods
**Nanopore membranes fabrication.** The $SiN_x$ and $MoS_2$ membranes are prepared using the previously reported procedure[13,37]. Briefly, 20-nm-thick supporting $SiN_x$ membranes are manufactured using anisotropic KOH etching to obtain 10 μm × 10 μm to 50 μm × 50 μm membranes, with size depending on the size of the backside opening. Reactive ion etching (RIE) is used to make a 50 nm opening on that membrane. CVD-grown $MoS_2$ flakes were transferred from sapphire substrates using the MoS2 transfer stage in a manner similar to the widely used graphene transfer method[9–11] and suspended on opening. Membranes are first imaged in the TEM with low magnification in order to check suspended $MoS_2$ flakes on opening.

Nanopores in $SiN_x$ membranes were made using a JEOL 2200FS high-resolution transmission electron microscope[7,8].

**DNA complex hybridization.** The following three oligonucleotides (named olig1, olig2, and olig3) were used to generate the ds–ss–ds DNA complex: >olig1

5′-CTTGGGAAATTGAATCACTTTTTGGTCCAGCAGTGAAATTTCCTG
TGACTCAGTAAATCACTTTAACTAATGAAGGAATAACAATCCCAGAGG
AGCAGAAGTTTCAACTATGCAGATTATTTCTGAGATTTAAAAAGTGAC
TCTTCAAAGAAATAAGTCCCTGGAGGCTGTTGGCTCTCTATAAAGGCTG
ACTTTCCACTCTCTTAAGTAGCTCATTTTGGCCCCGAGACAAGATTCAGT
TGGTGGTTTTTAATAAATAACGTTTTTGTATTACAAAAGTAAAATTTAGA
AAATTTACAGAAGAAGGAATTACGAAAAAACAACTCATTAGCCACAGA
CAACCAGATAATTCGTAATATGCCATTGTATTAATATTTCTTTCTGGTC
ATTTTTGATATGTCTGTTTTTATGTGATGCTAAACAAATTCAAATGCAA
TTGCATTGCTGCTTCA-3′ >olig2

5′-CTTTGAAGAGTCACTTTTTAAATCTCAGAAATAATCTGCATAGTT
GAAACTTCTGCTCCTCTGGGATTGTTATTCCTTCATTAGTTAAAGTGATT
TACTGAGTCACAGGAAATTTCACTGCTGGACCAAAAAGTGATTCAATTT
CCCAAG-3′ >olig3

5′-TGAAGCAGCAATGCAATTGCATTTGAATTTGTTTAGCATCACATAA
AAACAGACATATCAAAAATGACCAGAAAGAAATATTAATACAATGGCAT
ATTACGAATTATCTGGTTGTCTGTGGCTAATGAGTTGTTTTTTCGTAATT
CCTTC-3′

The sequences of olig1 and olig2 are reverse-complements of the sequences in bold in olig1. The three oligonucleotides can thus be annealed to form a ds–ss–ds DNA complex.

The protocol is detailed below: (1) mix three oligos in the ratio of 1:1:1 in the annealing buffer (100 mM potassium acetate; 30 mM HEPES, pH 7.5); (2) load the sample into an eppendorf tube and ramp quickly to 95 °C and hold for 30 min; (3) cool down to 75 °C slowly with 20 cycles (1 cycle/min) with the cycle step of 1 °C with a rate of 0.3 °C/s; (4) hold at 75 °C for 30 min; (5) cool down to 25 °C slowly with 50 cycles (1 cycle/min) with the cycle step of 1 °C with a rate of 0.3 °C/s; (6) hold at 4 °C until nanopore experiments or AFM measurements.

For short protruded complex, propynyl-modified cytosine oligos are used to enhance the affinity of a standard oligonucleotide sequence to its complementary nucleotide strand. Each modified base can raise the melting temperature up to 2.8 °C after hybridization. Generally, it can be used to label short DNA/RNA targets where normal oligonucleotides do not show sufficient affinity or specificity.

**AFM imaging.** AFM imaging was performed in photothermal off-resonance tapping[45] on a custom-built microscope based on a Bruker MultiMode AFM, as describe elsewhere[46]. DNA hybrids were diluted in 10 mM HEPES (4-(2-hydroxyethyl)-1-

piperazineethanesulfonic acid) pH 7.3 to a concentration of about 5 ng/μL. Two microliters of 20 mM $NiCl_2$ solution was pipetted on a freshly cleaved 3 mm Mica disk. Then, 2 μL of 10 mM HEPES pH 7.3 solution was added. Finally, 500 nL of the diluted DNA solution was added. The sample was left to incubate for several minutes to let the DNA adhere to the Mica. Afterwards the scan head with a hanging droplet of 10 mM HEPES supplemented with 2 mM $NiCl_2$ was added on top of the scanner for imaging. Images were recorded using Bruker FastScan-D probes at 30 kHz PORT rate and at 10 lines/s.

**Electrical recording**. Current–voltage, *IV*, characteristic and DNA translocation were recorded on an Axopatch 200B patch clamp amplifier (Molecular Devices, Inc. Sunnyvale, CA). We use a NI PXI-4461 card for data digitalization and custom-made LabView software for data acquisition using Axopatch 200B. The sampling rate is 100 kHz and a built-in low-pass filter at 10 kHz is used. Data analysis enabling event detection is performed offline using a custom open source Matlab code, named OpenNanopore[47] (http://lben.epfl.ch/page-79460-en.html).

For ECR nanopore making, a pair of chlorinated Ag/AgCl electrodes was employed to apply the transmembrane voltage and the current between the two electrodes was measured by a FEMTO DLPCA-200 amplifier (FEMTO® Messtechnik GmbH). A low voltage (100 mV) was applied to check the current leakage of the membrane. Details are published previously[37].

**Code availability**. All code used in this analysis is available in the Supplementary Software file.

**Reporting summary**. Further information on experimental design is available in the Nature Research Reporting Summary linked to this article.

## Data availability
The datasets generated and/or analyzed during the current study are available from the corresponding authors on reasonable request.

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

## Acknowledgements

This work was financially supported by Swiss National Science Foundation (Grant numbers 200021_153653, BSCGI0_157802, Roche SRA 1669 (to A.R.) and the DARPA Molecular Informatics Program under grant number KC# 090165-AF894 (to O.M.) as well as ERC Consolidator Grant 2017; InCell, Grant agreement no. 773091 (to G.E.F.). We thank the Centre Interdisciplinaire de Microscopie Électronique (CIME) at EPFL for access to their electron microscopes. Devices fabrication was partially carried out at the EPFL Center for Micro/Nanotechnology (CMi). We thank Andras Kis and Dumitru Dumcenco for kindly providing high-quality single-layer CVD MoS$_2$.

## Author contributions

K.L., A.K., O.M. and A.R. conceived the idea and designed all experiments. K.L. fabricated the nanopore devices and performed measurements. C.P. and O.M. performed the data analysis, model testing, and signal processing tasks. A.P.N. and G.E.F. contributed to AFM measurements. All authors wrote the manuscript and provided constructive comments.

## Additional information

**Competing interests:** The authors declare no competing interests.

