## [Peer Review File · Nature Communications]

Reviewers' Comments:

Reviewer #1:

Remarks to the Author:

Liu and co-worker's manuscript demonstrated that the solid-state nanopore can be used to detect topological variation in single DNA molecules. They applied ds-ss-ds DNA to explore the success rate and velocity profile of translocation events in SiNx nanopore. They claimed the high spatial resolution of 1 nm of the presented MoS2 nanopore. This is a very interesting paper and the experiments is well performed. However, the presented mechanism of DNA translocation seems plausible. The frequency analysis seems hardly supports the translocation mechanism. And the "1 nm spatial resolution" they claim should be confirmed by their experimental results. At this stage, I could not recommend this manuscript to be publish in Nature Communications without a major revision.

1.This is about the translocation events in SiNx nanopore. Authors used AFM to image the ds-ss-ds DNA hybrid molecule and reckoned that "the overall yield of forming ds-ss-ds complex is more than 80%". Then classify the spike-like signatures into '212', corresponding to complete translocation; '2' and '21', that was thought as partial translocation. And only 10% of the signatures were assigned to the '212'. Authors attributed this to a high entropic barrier for a complete translocation in such a small nanopore, but not explained clearly. How does the entropic barrier affect the translocation behavior? Why the complex translocates partially? And what about the rest part of the complex? Did the complex unzip under the interaction of nanopore? (Chem. Commun., 2017,53, 3539-3542) Authors need to give a more detailed explanation.

2.Authors concluded that "only a small fraction of spike-like event can be attributed to translocation" in abstract. How did authors get this conclusion? Can this small fraction be estimated?

3.Table 1 are not clear. In addition, b should have been β ?

4.In MoS2 experiment, author thought a salt gradient condition can "facilitate the capture of DNA segments and elongate the translocation time". How did authors get this conclusion without controllable experiments? If it's from the literature, authors should add a reference identifier.

5.Under the action of an external electric field, the electrons of the semiconductors, such as SiNx membrane, will move in a direction to form a current. For MoS2, there is also leakage current under applied voltage, which can be seen in Figure S4. Can authors explain the resource of the leakage current?

6.Authors need a uniform format, such as MoS2 should be MoS₂ in page 14.

7.In Page 6, the author state that "the overall yield of forming ds-ss-ds complex is more than 80%" which is based on their proposed translocation mechanisms of ds-ss-ds complex. The author should use other characterization methods to determine the yield of ds-ss-ds complex.

8.The author ascribes the "212" type of signature to the real translocation the ds-ss-ds complex. However, the entrance of first ds-ss part of DNA follows by a bump out behavior could also contribute to "212" type of events. The author should design an asymmetric ds-ss-ds complex to confirm the real translocation signature.

9.Previous studies on biological nanopores have already show the "partial" translocation through the nanopore (Nat. Nanotechnol., 11, 713-716, 2016, Chem. Commun. 48, 8784-8786, 2012). Please discuss the results presented in this manuscript with the previous studies on biological nanopores.

10.As shown in figure 2b, the "212" events own the distinguishable two level current with large current difference at nA level. However, the author further uses the noise analysis to "discriminate" these "212" events. What's the advantages of frequency analysis for discriminating of "212" events? What can we learn from the frequency noise analysis? For example, is there any new knowledge could be gained for the translocation process of DNA complex? the interactions between pore and analyte? Since the nanopore current traces is non-stationary and non-linear, it is improper to use the PSD analysis to gain the detail frequency information, see Faraday Discussions, DOI: 10.1039/C8FD00023A (2018)

11.The author demonstrates that they have achieve 1 nm spatial resolution. If it is the real 1 nm

spatial resolution, this presented nanopore should discriminate 1 nm difference along the whole length of DNA not only the barcode part. On the other hand, the author should consider the conformational changes of the DNA complex induced by the barcode when they propose the "1 nm spatial resolution".

12. The previous nanopore study reported the 0.07 nm³ resolution for reading the primary structure of a protein (Nature Nanotechnology, 2016, 10.1023/NNANO.2016.120). The author should cite this paper and compare their spatial resolution to this paper.

Reviewer #2:

Remarks to the Author:

The manuscript by Liu et al describes the use of SiNx and MoS₂ pores for distinguishing ssDNA from dsDNA in a passing DNA strand. While this is an important step in barcoding DNA, there are several issues that raise concerns and these should be addressed in full prior to consideration for publication. Some of these are minor and suggestive, and they are not organized in any particular order of importance, but all should be addressed if possible, because the remaining points question the reproducibility and robustness of what is shown in the manuscript.

- p-12: electrodes is mis-spelled
- authors refer to distance between bases to be ~0.3nm, however, in ssDNA this distance is larger (~0.5nm), because ssDNA does not have a helical structure.
- p-13: it is not clear why oxide protruding from MoS₂ pore would slow down DNA. Can the authors develop this idea, or provide a reference that explains this?
- on many occasions the authors refer to hemolysin as hemolysins, I wonder what the reason is?
- the AFM image in S2 looks weird, the scan lines are at a strange angle, can the authors perhaps show a normal scan to convey their point? The 3D representation looks not convincing and artifact-full.
- on page 7, the authors say that the issue of entropic barrier in solid-state pores has rarely been discussed, however, there are several papers that discuss this with silicon nitride pores, including Wanunu 2008 Biophysical J. and van den Hout Biophysical J. 2010 to name a couple. Also, Carson et al (Biophysical J 2015) did a very systematic study of DNA going through pores in nitride pores of the same approximate geometry as in this paper.
- SI, p-11: "As a result, we are able to...?"
- What is the yield of the DNA constructs? It would be valuable to see gels.
- the authors do not show any continuous traces, to allow one to see the quality of the pores.
- in the events shown in Figure 2, the open pore levels before and after most events are substantially different, as indicated by the raw (blue) trace as well as the fit (red). Why is that the case?
- The barcode data with MoS₂ pore is not too convincing, especially since very few datapoints are shown as traces, in contrast to the large number of points in the scatter plots. In 2 of the 7 events shown, the spike corresponding to dsDNA looks like noise, which goes both up and down for the events.
- The barcode current level of the histogram in the figure does not have a distinct characteristic peak, but rather looks like the tail of the ssDNA peak.
- Have any of these results been repeated for multiple pores? If so, what is the pore-to-pore variation?

Reviewer #1 (Remarks to the Author):

Liu and co-worker's manuscript demonstrated that the solid-state nanopore can be used to detect topological variation in single DNA molecules. They applied ds-ss-ds DNA to explore the success rate and velocity profile of translocation events in SiNx nanopore. They claimed the high spatial resolution of 1 nm of the presented MoS₂ nanopore. This is a very interesting paper and the experiments is well performed. However, the presented mechanism of DNA translocation seems plausible. The frequency analysis seems hardly supports the translocation mechanism. And the "1 nm spatial resolution" they claim should be confirmed by their experimental results. At this stage, I could not recommend this manuscript to be publish in Nature Communications without a major revision.

We thank the reviewer for acknowledging the importance of our work. Regarding the concerns about the translocation mechanism and the frequency analysis, we made detailed discussions in the revised version. Regarding the statement of "1 nm resolution", the aim of this paper is to provide a methodology to target specific short sequences instead of reading every single base in DNA. Due to the thin nature of MoS₂ membranes, we are able to detect 1 nm motif attached to ssDNA. As we state, in the abstract "In addition to their use in DNA sequencing, ultrathin nanopore membranes have potential applications in detecting topological variations in DNA", this paper gives a roadmap beyond sequencing.

1. This is about the translocation events in SiNx nanopore. Authors used AFM to image the ds-ss-ds DNA hybrid molecule and reckoned that "the overall yield of forming ds-ss-ds complex is more than 80%". Then classify the spike-like signatures into '212', corresponding to complete translocation; '2' and '21', that was thought as partial translocation. And only 10% of the signatures were assigned to the '212'. Authors attributed this to a high entropic barrier for a complete translocation in such a small nanopore, but not explained clearly. How does the entropic barrier affect the translocation behavior? Why the complex translocates partially? And what about the rest part of the complex? Did the complex unzip under the interaction of nanopore? (Chem. Commun., 2017,53, 3539-3542) Authors need to give a more detailed explanation.

We further implemented the "Kramer" barrier crossing model to elaborate the scenario, where the entropic barrier stemmed from the small nanopore used plays a vital role. The probability of a complete translocation has an exponential dependence on the barrier. We also used a similar barrier model for understanding molecular transport in nanopore in our previous publication (Nat. Nanotechnol., 10, 1070, 2015). The pore entrance can reject the rest of the complex. The unzipping can only happen when the pore diameter is smaller than dsDNA as reported in the above paper (Chem. Commun., 2017,53, 3539-3542). In our case, the pore diameter is larger than dsDNA. To clarify the

"In contrast, for narrow nanopores (~ 3 nm), the entropic barrier plays a vital role. We used a barrier model for understanding molecular transport cross a nanopore in our previous

publication.¹⁶ The probability of a complete translocation has an exponential dependence on the barrier energy.”

2. Authors concluded that “only a small fraction of spike-like event can be attributed to translocation” in abstract. How did authors get this conclusion? Can this small fraction be estimated?

Based on the fact that, from AFM measurements, we got high percentage of “212” complex as input materials for translocation experiments. However, we only observed a rather low percentage (approximately 10%) of “212” events.

3. Table 1 are not clear. In addition, b should have been β ?

We changed the typo in the revised version.

4. In MoS₂ experiment, author thought a salt gradient condition can “facilitate the capture of DNA segments and elongate the translocation time”. How did authors get this conclusion without controllable experiments? If it’s from the literature, authors should add a reference identifier.

We cited the relevant literature (Nat. Nanotechnol., 5, 160, 2010) in the revised revision, where a salt gradient was used to increase the capture rate.

5. Under the action of an external electric field, the electrons of the semiconductors, such as SiN_x membrane, will move in a direction to form a current. For MoS₂, there is also leakage current under applied voltage, which can be seen in Figure S4. Can authors explain the resource of the leakage current?

The low level of the leakage current can originate either from the defects on MoS₂ or the interface between SiN_x and MoS₂. The leakage current is usually a good indicator that electrochemical reaction (ECR)(REF Nano Letters) can happen for the lower transmembrane voltage range (0.8-1.5V). In the absence of the defects ECR threshold increases.

6. Authors need a uniform format, such as MoS₂ should be MoS₂ in page 14.

We uniformized abbreviation for molybdenum disulfide it in the revised version.

7. In Page 6, the author state that “the overall yield of forming ds-ss-ds complex is more than 80%” which is based on their proposed translocation mechanisms of ds-ss-ds complex. The author should use other characterization methods to determine the yield of ds-ss-ds complex.

The overall yield is deduced from AFM measurements. AFM gives a direct statistic based on counting single molecules. We didn’t perform bulk analysis, such as gel electrophoresis as the final concentration of our product was low (~ 1 nM).

Revised text is " The overall yield of forming ds-ss-ds complex is more than 80%, a result obtained after analyzing a number of such scans. AFM gives a direct statistic based on counting single molecules."

8.The author ascribes the "212" type of signature to the real translocation the ds-ss-ds complex. However, the entrance of first ds-ss part of DNA follows by a bump out behavior could also contribute to "212" type of events. The author should design an asymmetric ds-ss-ds complex to confirm the real translocation signature.

We used a "212121" design to prove the real translocation. As illustrated below, an asymmetric feature has been observed.

Figure R1. A "212121" event detected.

9.Previous studies on biological nanopores have already show the "partial" translocation through the nanopore (Nat. Nanotechnol., 11, 713-716, 2016, Chem. Commun. 48, 8784-8786, 2012). Please discuss the results presented in this manuscript with the previous studies on biological nanopores.

We discussed the previous studies as mentioned by the referee and we have modified the revised version to include this suggestion by adding the following clarification.

Revised text is "Partial translocatons have also been observed in biological nanopores,24-26 as well in the context of narrow solid-state nanopores.27,28 However, most of the solid-state nanopores are relatively big, facilitating complete translocation."

10.As shown in figure 2b, the "212" events own the distinguishable two level current with large current difference at nA level. However, the author further uses the nosie analysis to "discriminate" these "212" events. What's the advantages of frequency analysis for discriminating of "212" events? What can we learn from the frequency noise analysis? For example, is there any new knowledge could be gained for the translocation process of DNA complex? the interactions between pore and analyte? Since the nanopore current traces is non-

stationary and non-linear, it is improper to use the PSD analysis to gain the detail frequency information, see Faraday Discussions, DOI: 10.1039/C8FD00023A (2018)

The aim of our frequency analysis is to provide a precise means to characterize nanopore noise at different current levels and for different DNA topologies. It is correct that some “212” events have visually distinguishable current differences at the nA level, but many of them do not. In order to automatically detect “212” events, we developed two new detection methods, operating in the time and frequency domain.

Regarding the comment “Since the nanopore current traces is non-stationary and non-linear, it is improper to use the PSD analysis to gain the detail frequency information, see Faraday Discussions” we agree with the reviewer that there is reason to believe that the traces are nonstationary and nonlinear, but the reviewer should be aware that we are not trying to detect/estimate the bases, just changes in the topological structure -- this makes the problem analytically more tractable. The Hilbert transform method outlined in the 2018 work suggested by the reviewer performs poorly on our datasets, and we provided a number of examples to show that this is the case. Frequency analysis (i.e., PSD analysis) is the most commonly used tool in the field, and unlike the Hilbert transform method, it has analytical performance guarantees.

We have added detailed discussions in the revised main text and SI (section S3.4).

“We conclude the analysis by pointing out that a recent trend in nanopore signal processing is to use the Hilbert–Huang Transform (HHT)³³. HHT is a useful empirical tool for analyzing nanopore traces as it is designed to work for signals that are nonstationary and nonlinear. Still, the HHT method may not be useful for topological signal detection, as the information-bearing signal in this setting has significantly simpler statistical properties than the one encountered when performing DNA bases detection or estimation. To illustrate the point, we performed the HHT analysis, and one of the findings is shown in Figure SX. As may be seen, it is extremely hard to classify “2” and “1” fragments based on the HHT. A detailed discussion is relegated to SI.”

11. The author demonstrates that they have achieved 1 nm spatial resolution. If it is the real 1 nm spatial resolution, this presented nanopore should discriminate 1 nm difference along the whole length of DNA not only the barcode part. On the other hand, the author should consider the conformational changes of the DNA complex induced by the barcode when they propose the “1 nm spatial resolution”.

We demonstrated that 1 nm protruded feature along ssDNA can be distinguished by nanopore sensing. As we stated in the abstract, our aim is not to sequence the bases along the ssDNA, but to provide a barcode method to detect specific part along the strand.

12. The previous nanopore study reported the 0.07 nm³ resolution for reading the primary structure of a protein (Nature Nanotechnology, 2016, 10.1023/NNANO.2016.120). The author should cite this paper and compare their spatial resolution to this paper.

We would like to thank the reviewer for this valuable suggestion, however, given the focus of our paper on the barcoding, we don't find it fitting.

Reviewer #2 (Remarks to the Author):

The manuscript by Liu et al describes the use of SiNx and MoS₂ pores for distinguishing ssDNA from dsDNA in a passing DNA strand. While this is an important step in barcoding DNA, there are several issues that raise concerns and these should be addressed in full prior to consideration for publication. Some of these are minor and suggestive, and they are not organized in any particular order of importance, but all should be addressed if possible, because the remaining points question the reproducibility and robustness of what is shown in the manuscript.

We thank the reviewer for acknowledging the importance of our work. We addressed his useful comments below.

- p-12: electrodes is misspelled

Thank you for noticing a spelling mistake, in the revised version we corrected it.

- authors refer to distance between bases to be ~0.3nm, however, in ssDNA this distance is larger (~0.5nm), because ssDNA does not have a helical structure.

We rephrased in the revised version to be more precise.

“equal to twice the distance between two bases in dsDNA”

- p-13: it is not clear why oxide protruding from MoS₂ pore would slow down DNA. Can the authors develop this idea, or provide a reference that explains this?

Here is the theoretical study that shows the edge chemistry can affect the translocation behavior (ACS Nano, 2014, 8, pp 7914–7922). MoS₂ nanopore benefits from a craftable pore architecture (combination of Mo and S atoms at the edge). Functionalization on the pore edge has also been proposed to slow down the transport of DNA.

Revised text is “Electrochemically etched MoS₂ nanopores have oxide protruding around the pore fringe, which has various possibilities of functionalization, and thus can further slow down the translocation.”

- on many occasions the authors refer to hemolysin as hemolysins, I wonder what the reason is?

We made the correction. It should be hemolysin.

- the AFM image in S2 looks weird, the scan lines are at a strange angle, can the authors perhaps show a normal scan to convey their point? The 3D representation looks not convincing and artifact-full.

We agree with the reviewer that the 3D representation doesn't give added information. We removed it for S2 to avoid misunderstanding.

- on page 7, the authors say that the issue of entropic barrier in solid-state pores has rarely been discussed, however, there are several papers that discuss this with silicon nitride pores, including Wanunu 2008 Biophysical J. and van den Hout Biophysical J. 2010 to name a couple. Also, Carson et al (Biophysical J 2015) did a very systematic study of DNA going through pores in nitride pores of the same approximate geometry as in this paper.

Thanks for pointing out the issue. We cited the relevant papers in the revised version.

As a result, we changed the phrasing to be:

"The partial translocations have also been observed in biological nanopores,²⁴⁻²⁶ as well in the context of narrow solid-state nanopores.^{27,28} However, the phenomenon is less frequently observed in the most solid-state nanopores, as their sizes were relatively large compared to biological pores."

-SI, p-11: "As a result, we are able to...?"

It should be "The fitted parameters are listed in Table 1."

- What is the yield of the DNA constructs? It would be valuable to see gels.

AFM gives a direct statistic based on counting single molecules. The amount of the final complex is very low (~ 1 nM) to run a gel experiment.

- the authors do not show any continuous traces, to allow one to see the quality of the pores.

We added a continuous trace to show the pore condition.

Figure R2. A 2.5 second continuous trace low-pass filtered at 10 kHz.

- in the events shown in Figure 2, the open pore levels before and after most events are substantially different, as indicated by the raw (blue) trace as well as the fit (red). Why is that the case?

This is due to the local environment of ions affected by the translocation. It is noted that the very few data points after one event show very large fluctuation, especially in small nanopores. As shown in this work (Science. 2010; 327: 64–67.), this is probably due to the relaxation of ions inside the nanopore after translocation. At a longer timescale, for example, as shown in Figure 3a, the baseline is generally constant.

- The barcode data with MoS₂ pore is not too convincing, especially since very few datapoints are shown as traces, in contrast to the large number of points in the scatter plots. In 2 of the 7 events shown, the spike corresponding to dsDNA looks like noise, which goes both up and down for the events.

The complex with ds part has a significantly longer dwell time compared to ss-DNA. We speculate that the barcoded ds part has a specific interaction with the nanopore edge, which may slow down the speed. Moreover, as shown in Figure 4e, two entering signatures have been experimentally observed. This, indicates that the secondary-level blockage indeed originates from the topological change along the DNA strand. In the control experiments, we have translocated ssDNA and no 2 level events were observed. Regarding the noise, as we discussed for ds-ss-ds complex, a change in the topology can affect the noise features significantly.

- The barcode current level of the histogram in the figure does not have a distinct characteristic peak, but rather looks like the tail of the ssDNA peak.

All the ssDNA current values are lower than 2 nA. And dsDNA level has much fewer data points compared to ssDNA level and openpore level. However, three distinct levels are distinguishable in the histogram.

- Have any of these results been repeated for multiple pores? If so, what is the pore-to-pore variation?

Yes, figure 2 and figure 3 are from different pores, which have different current levels. “212” features have been observed in different pores. Different pores may have different entropic barriers, thus affecting the distribution of clusters.

Figure R3. DNA complex in two nanopores. a) a 3 nm nanopore recorded at 400 mV. ~ 3000 events are collected. b) a 5 nm nanopore recorded at 400 mV. ~ 2000 events are collected. For both plots, the time scale is set at logarithmic scale (ms).

Reviewers' Comments:

Reviewer #1:

Remarks to the Author:

The authors have addressed most of my comments. However, there are still several concerns need to be clarified before the final acceptance.

I. There are some incomplete sentence and ambiguous statement in the response letter. See as follows:

1. response to question 1 of Reviewer 1: "To clarify the..."
2. response to question 5 of Reviewer 1 "The leakage current is usually a good indicator that electrochemical reaction (ECR)(REF Nano Letters)can happen for the lower transmembrane voltage range (0.8- 1.5V). In the absence of the defects ECR threshold increases."

II. The reviewer 2 concerned the reproducibility of the experiments. The author provides a continuous trace with total recording time of 2.5 s as a Figure R2. I'm wondering is this figure from the SiNx nanopore or MoS2 nanopore? What's the lifetime of the MoS2 and the SiNx nanopore, respectively. Is it possible for the author to include the continuous trace into their supplementary information? There are obvious fluctuations in the baseline of Figure R2. Would this fluctuation affect the reproducibility of this methods since the surface properties of each pore may different? Therefore, the signature of the characteristic event may varied from pore-to-pores.

III. The supplementary figure R3 for the multiple pores is from the SiNx pores? The author should indicate the experimental condition of the figure. The pore diameter seems affect the sensing ability of this proposed technique. What's the difference of the sensing ability in discrimination of topological variations of DNA as the pore diameter varied? How about the reproducibility of MoS2 nanopore?

IV. The author claimed in question 10 in Reviewer 1 that "Still, the HHT method may not be useful for topological signal detection, as the information-bearing signal in this setting has significantly simpler statistical properties than the one encountered when performing DNA bases detection or estimation." This conclusion is incorrect. The HHT provides the characteristic frequency information including the frequency band, frequency energy and the transient variations for a single molecule inside the nanopore. The Figure S3 seems plausible. According to their Main text (Figure 2a), the duration time of Level 0, 1, and 2 in one signature last several hundred microseconds. However, in Figure S3, the events for each level last very long time (the author did not provide the time unite). According to their sampling rate of 100 kHz, the signal in Figure S3 seems have 12 – 35 ms which is significantly longer than the duration time of Level 0, 1, and 2 in Figure 2a. One possible procedure the author may adopted is that they connected the durations from many signals before perform the HHT. Since the duration splicing would induced the significant variation in frequency analysis, the author may not have got the proper HHT results. Moreover, there are significant different in IMF 1 and 2, the HAS analysis should distinguish Level 0, 1, and 2. The author provides Figure S4 as statistical results but not indicate the frequency range. In other words, Figure S4 may not the proper frequency range for recognizing Level 0, 1, and 2. The PSD analysis may have the performance in this case, but not the self-adapting methods for the non-stationary and non-linear nanopore data.

Reviewer #2:

Remarks to the Author:

The authors have done a good job of addressing most of the comments, especially with revising the AFM images and the text. However, there are still a few that need addressing upon a read of their revised version, and I would support publication of the work after these points are

addressed:

1) Page 3: "Second, the incorporation of a so-called "molecular motor" onto the pore mouth can ratchet single strand DNA in a base-by-base manner. However, this also limits the sequencing speed to the range of 5-20 nt/second. " I think a reference is needed here, and I believe the actual rate is more like 500 nt/sec, not 5-20.

2) Page 6: "The current blockages induced by ssDNA and dsDNA are 0.8 nA and 1.5 nA, respectively." Can the authors add st. dev. values for these numbers, and also indicate the voltage?

3) In Figure 2b, two of the events shown are identical (both are 212 type). The authors should revise that figure to include a new event (given that there are so many events in the scatter plot, a few more would be nice to show).

We want to thank reviewers for their useful comments and suggestions.

Reviewer #1 (Remarks to the Author):

The authors have addressed most of my comments. However, there are still several concerns need to be clarified before the final acceptance.

I. There are some incomplete sentence and ambiguous statement in the response letter. See as follows:

1. response to question 1 of Reviewer 1: “To clarify the...”

It should be “To clarify, we revised the text...”. We apologize for deletion of the text.

2. response to question 5 of Reviewer 1 “The leakage current is usually a good indicator that electrochemical reaction (ECR)(REF Nano Letters)can happen for the lower transmembrane voltage range (0.8- 1.5V). In the absence of the defects ECR threshold increases.”

The reference should be “Nano Lett., 2015,15, 3431-3438”, which is the ref. 36 in the manuscript. We apologize for the ambiguity.

II. The reviewer 2 concerned the reproducibility of the experiments. The author provides a continuous trace with total recording time of 2.5 s as a Figure R2. I’m wondering is this figure from the SiN_x nanopore or MoS₂ nanopore? What’s the lifetime of the MoS₂ and the SiN_x nanopore, respectively. Is it possible for the author to include the continuous trace into their supplementary information? There are obvious fluctuations in the baseline of Figure R2. Would this fluctuation affect the reproducibility of this methods since the surface properties of each pore may different? Therefore, the signature of the characteristic event may varied from pore-to-pores.

Figure R2 is the trace from a SiN_x nanopore. In general, the lifetime of a typical MoS₂ nanopore is 2-10 hrs. While for a SiN_x nanopore it is typically more than 10 hrs But normally, 2-3 hrs will give sufficient data for the statistical analysis. We added continuous traces for two types of pores in SI.

The fluctuation in the baseline wouldn’t affect the reproducibility of the method described since we only calculated short fragment of baseline before each event for event detection.

The absolute value of each level in “212” events may vary from pore to pore. But the key feature for such events will remain the same according to our observation using many different pores.

III. The supplementary figure R3 for the multiple pores is from the SiN_x pores? The author should indicate the experimental condition of the figure. The pore diameter seems affect the sensing ability of this proposed technique. What’s the difference of the sensing

ability in discrimination of topological variations of DNA as the pore diameter varied? How about the reproducibility of MoS₂ nanopore?

Yes, it is from two SiN_x pores. The experimental conditions are the same as written in the main text. According to our observations, smaller pore will lead to longer dwell time, which facilitates the multi-level detection. However, it also lowers the probability of observing “212” due to its high entropic barrier.

We are able to reproduce the results in two MoS₂ nanopores. As shown in Figure R1, the absolute value of each level can be different from pore to pore, and the secondary blockage can be clearly observed.

Figure R1. Translocation of hybrid DNA in a 5 nm MoS₂ nanopore.

IV. The author claimed in question 10 in Reviewer 1 that “Still, the HHT method may not be useful for topological signal detection, as the information-bearing signal in this setting has significantly simpler statistical properties than the one encountered when performing DNA bases detection or estimation.” This conclusion is incorrect. The HHT provides the characteristic frequency information including the frequency band, frequency energy and the transient variations for a single molecule inside the nanopore. The Figure S3 seems plausible. According to their Main text (Figure 2a), the duration time of Level 0, 1, and 2 in one signature last several hundred microseconds. However, in Figure S3, the events for each level last very long time (the author did not provide the time unite). According to their sampling rate of 100 kHz, the signal in Figure S3 seems have 12 – 35 ms which is significantly longer than the duration time of Level 0, 1, and 2 in Figure 2a. One possible procedure the author may adopted is that they connected the durations from many signals before perform the HHT. Since the duration splicing would induced the significant variation in frequency analysis, the author may not have got the proper HHT results. Moreover, there are significant differences in IMF 1 and 2, the HAS analysis should distinguish Level 0, 1, and 2. The author provides Figure S4 as statistical results but not

indicate the frequency range. In other words, Figure S4 may not be the proper frequency range for recognizing Level 0, 1, and 2. The PSD analysis may have the performance in this case, but not the self-adapting methods for the non-stationary and non-linear nanopore data.

Our apology, we indeed used the concatenated signal for obtaining the results plotted in Figure S3. Thank you for pointing this out. We now added a proper explanation of how the IMFs are computed without concatenation and changed Figure S3 accordingly. The time duration of events 1 and 2 lies typically in the range 1-4ms for a small nanopore, while the duration of event 0 is set around 10ms. As may be seen from the figures, the IMFs are now significantly smoother, and distinguishable for different levels. With regards to Figure S4, we believe that the frequency range was included in the original plot – the caption of the figure reads as “Frequency: 0-10K”. One thing we added in the revised manuscript is an explanation that reads as follows “Note that level 1 signals tend to have larger energy in the given frequency range than level 0 and 2 signals, and energy may be consequently used as a means to discriminate the events. Better results may be expected for a larger number of events, as our analysis only makes use of 40 samples”. We also added a sentence, as suggested by the reviewer, indicating that “PSD analysis is suitable for topological signal detection, as the information-bearing signal in the studied setting has significantly simpler spectral discriminators than the one encountered when performing DNA bases detection or estimation. Still, HHT remains a useful tool that self-adapts to nonstationary and nonlinear nanopore signals.”

Reviewer #2 (Remarks to the Author):

The authors have done a good job of addressing most of the comments, especially with revising the AFM images and the text. However, there are still a few that need addressing upon a read of their revised version, and I would support publication of the work after these points are addressed:

1) Page 3: "Second, the incorporation of a so-called “molecular motor” onto the pore mouth can ratchet single strand DNA in a base-by-base manner. However, this also limits the sequencing speed to the range of 5-20 nt/second. " I think a reference is needed here, and I believe the actual rate is more like 500 nt/sec, not 5-20.

Thanks for pointing out the speed issue. We added an appropriate reference (Nat. Biotechnol., 2018, 36, 338) to cite this number (450 nt/second).

2) Page 6: "The current blockages induced by ssDNA and dsDNA are 0.8 nA and 1.5 nA, respectively." Can the authors add st. dev. values for these numbers, and also indicate the voltage?

We revised the text and added st.dev. values.

3) In Figure 2b, two of the events shown are identical (both are 212 type). The authors

should revise that figure to include a new event (given that there are so many events in the scatter plot, a few more would be nice to show).

We added a new event in the revised Figure 2.

Reviewers' Comments:

Reviewer #1:

Remarks to the Author:

The author has thoroughly response to my concerns. Their solid-state nanopores could have the reproducibility for the shape of characteristic signals. However, it is better to discuss the pore-to-pore difference in the manuscript and even provide potential solution. This part will properly guide the readers to understand their strategy and the novelty of this manuscript. At this stage, I would like to recommend this manuscript to be published after a minor revision.

REVIEWERS' COMMENTS:

Reviewer #1 (Remarks to the Author):

The author has thoroughly response to my concerns. Their solid-state nanopores could have the reproducibility for the shape of characteristic signals. However, it is better to discuss the pore-to-pore difference in the manuscript and even provide potential solution. This part will properly guide the readers to understand their strategy and the novelty of this manuscript. At this stage, I would like to recommend this manuscript to be published after a minor revision.

Thanks the referee for acknowledging our work. I added the corresponding discussion into the manuscript as following “One technical obstacle is the variation in nanopore fabrication. We observe large pore-to-pore difference in dwell time and current. Noise analysis and pattern recognition could be possible solutions to increase the fidelity of the method reported. “